# GL-Tree: A Hierarchical Tree Structure for Efficient Retrieval of Massive Geographic Locations

**DOI:** 10.3390/s23042245

**Published:** 2023-02-16

**Authors:** Bin Liu, Chunyong Zhang, Yang Xin

**Affiliations:** 1National Engineering Laboratory for Disaster Backup and Recovery, Information Security Center, School of Cyberspace Security, Beijing University of Posts and Telecommunications, Beijing 100876, China; 2College of Information and Network Engineering, Anhui Science and Technology University, Bengbu 233030, China

**Keywords:** location-based services, location privacy protection, Geohash code, location privacy space index

## Abstract

Location-based application services and location privacy protection solutions are often required for the storage, management, and efficient retrieval of large amounts of geolocation data for specific locations or location intervals. We design a hierarchical tree-like organization structure, GL-Tree, which enables the storage, management, and retrieval of massive location data and satisfies the user’s location-hiding requirements. We first use Geohash encoding to convert the two-dimensional geospatial coordinates of locations into one-dimensional strings and construct the GL-Tree based on the Geohash encoding principle. We gradually reduce the location intervals by extending the length of the Geohash code to achieve geospatial grid division and spatial approximation of user locations. The hierarchical tree structure of GL-Tree reflects the correspondence between Geohash codes and geographic intervals. Users and their location relationships are recorded in the leaf nodes at each level of the hierarchical GL-Tree. In top–down order, along the GL-Tree, efficient storage and retrieval of location sets for specified locations and specified intervals can be achieved. We conducted experimental tests on the Gowalla public dataset and compared the performance of the B+ tree, R tree, and GL-Tree in terms of time consumption in three aspects: tree construction, location insertion, and location retrieval, and the results show that GL-Tree has good performance in terms of time consumption.

## 1. Introduction

### 1.1. Overview

With the continuous development of mobile network technology, the number of mobile users continues to grow. Application services based on mobile technology continue to seek new technological innovations and economic growth points. The security threats faced by mobile networks have aroused widespread concern. Ensuring mobile network communication security, and that sensitive information does not fall into the hands of attackers, has become an increasingly complex problem.

The main security threats affecting mobile network applications are divided into several categories. This includes data leakage, social engineering attacks [1], jamming attacks [2], software virus security, cryptojacking attacks [3], wireless channel security, physical device violations, and user location privacy leakage. Researchers have conducted research on these security issues from security threat classification [4], security defense scheme design, security performance optimization, evaluation, etc. The research covers access domain security, network domain security, user domain security, and application domain security. Among them, access domain security research mainly involves access control, user positioning [5], security authentication [6], jamming attack detection and traceability [2], etc. Network domain security research mainly involves network transmission encryption and decryption [7,8], abnormal traffic control [9,10], signaling and protocol filtering, attack defense and traceability analysis, transmission security assessment, etc. User domain security research includes user behavior analysis [11], user privacy protection [12,13,14], etc. Application domain security involves access control, code security, application security awareness, vulnerability mining, and content security.

Location-based service (LBS) is a typical mobile application service. It uses precise location positioning technology and mobile network communication technology to provide users with geolocation-based information services based on their query needs. Such typical services include point-of-interest queries, location sharing, nearest-neighbor queries, mobile navigation, fast food ordering, etc. While LBS provides convenient services for people, it also introduces the risk of personal information security. A malicious attacker can combine background knowledge after obtaining the user’s location dataset to mine the specific users’ privacy information, such as location information, motion trajectory, personal identity, marital status, habits and hobbies, workplace, home address, health status, political beliefs, family members, etc. Researchers have conducted numerous studies on mobile user location privacy.

From a global perspective, due to many factors such as geography, economic and social development status, people’s living customs, religion, and culture, it is very unevenly distributed in space and time that locations are collected by various types of location-based service applications. Therefore, when engaging in application development and scientific research of location-based datasets, how to effectively manage, store, and efficiently retrieve location data in the face of the massive and unevenly distributed location data becomes a critical problem to be studied and solved.

### 1.2. Research Objectives and Motivation

The centralized architecture has been widely adopted in the study of location privacy and trajectory privacy protection schemes due to its structural simplicity, low communication cost, and good privacy protection capabilities. This architecture introduces a trusted third-party anonymous server between user terminals and LSPs (location service provider servers). When a user requests location services, the anonymous server receives the user’s location service query request, implements a location privacy protection algorithm, constructs an anonymous location query set, and forwards the anonymously processed query request to the LSPs. When the LSPs finish responding to the location service query request, the anonymous server receives the query response result set, filters the response results, and forwards the user’s desired query results into the user terminal. Thus, the anonymous server blocks direct communication between users and untrustworthy LSPs, relays query requests and query responses, and implements the location privacy protection algorithm.

When constructing anonymous location query sets, algorithms typically use fake locations, other cooperating users’ current locations, or historical locations. Algorithms that use fake locations introduce randomness that may lead to the unreasonable geographic distribution of locations. An attacker with background knowledge can identify fake locations with a higher probability. The algorithm using the current locations of other cooperative users needs to introduce more communication overhead and also suffers from the problem that the number of cooperative users cannot satisfy the k-anonymity requirement. Constructing anonymous location sets based on a massive number of historical locations solves the above issues. However, facing the storage and processing of massive historical location data, the anonymous server may become a bottleneck affecting the performance of the whole LBS system.

The motivation of this paper is to investigate the characteristics of Geohash encoding and design a data structure to address the possible bottleneck in the performance of centralized privacy protection structures. This data structure utilizes Geohash encoding to regionalize and fuzzify the user’s exact location for location hiding, improves the performance of the anonymization server for storing and retrieving large amounts of location data, and reduces the execution time of the anonymization algorithm.

### 1.3. Proposal Overview

In this paper, in the process of conducting research on location privacy protection, based on the analysis of the principle and characteristics of Geohash encoding, in order to achieve efficient retrieval of user locations from massive location datasets and to combine the effect of Geohash coding length variation on location intervals, we design and develop GL-Tree, a hierarchical tree structure for the storage and efficient retrieval of massive user locations. We elaborate on the structure, physical meaning, location storage and retrieval process, as well as algorithm pseudo-code implementation of GL-Tree. We conduct experiments on the Gowalla public dataset to verify the performance of GL-Tree.

### 1.4. Paper Outline

The rest of this paper is organized as follows. Section 2 discusses related work. Section 3 describes the principles, pseudo-code, and feature analysis of Geohash. Section 4 describes in detail the structure, physical meaning, and location storage and retrieval process of GL-Tree. Section 5 elaborates on and analyzes the related algorithms in GL-Tree. Section 6 picks up on the Gowalla public dataset and experimentally compares and analyzes GL-Tree with the B+ tree and R tree in terms of time performance of tree structure creation, location insertion, and retrieval. Section 7 concludes the full text.

## 2. Related Work

### 2.1. Research Related to Location Privacy Protection

The research on location privacy protection mainly includes location-based privacy protection and trajectory-based privacy protection. Data encryption technology is one of the most commonly used methods to protect sensitive information and prevent unauthorized use [8]. However, due to the complexity of encryption algorithm operations and resource consumption, it is currently less used in the design of location privacy protection schemes. Currently, the main methods proposed by researchers to protect location privacy and trajectory privacy are dummies, mix-zones, location perturbation and obfuscation, spatial cloaking, temporal generalization, release suppression, privacy information retrieval, and differential privacy. The design idea of these methods is mainly to achieve privacy protection by increasing the uncertainty of the attacker inferring the exact location of the user. Many methods combine *k*-anonymity privacy protection ideas.

*K*-anonymity was first applied to the design of location privacy protection by Gruteser [12]. The scheme uses the user’s exact location and other k−1 locations together to construct a location anonymization set. Query requests are sent to LSPs as anonymous sets so that the probability of LSPs identifying the user’s exact location does not exceed $\frac{1}{k}$. Therefore, *k*, also known as anonymity, is an important measure of the algorithm’s privacy-preserving capability. The larger the value of *k*, the better the privacy protection.

Since Gruteser’s study assumes that all users have the same *k*-value, it cannot provide the need for personalized anonymity. Gedik [15] proposes the CliqueCloak scheme, which meets the need for personalized anonymity, maximum execution time, and spatial tolerance. Mokbel [16] proposed the Casper scheme, which uses a location anonymizer to hide the user’s location in a spatial region. The hidden space is constructed based on the pyramidal data structure of the grid.

Kido [17] first proposed a location privacy-preserving scheme using dummy locations to achieve *k*-anonymity, where the user submits the exact location and a series of generated dummy locations directly to LSPs. Lu et al. [18] proposed two-dummy generation algorithms, CirDummy and GridDummy. The algorithms divide the user’s area into several equal-sized sub-regions so that the user’s location and the randomly generated dummy locations are evenly distributed in these sub-regions to increase the difficulty of guessing. However, the randomly generated dummy location scheme, in turn, may produce unreasonable dummy locations (e.g., rivers, lakes, swamps, mountains, etc.). The attacker can easily exclude the dummy locations from the probabilistic point of view, thus reducing the privacy-preserving capability of the scheme.

To address the problem that adversaries may use background knowledge to reduce the privacy of *k*-anonymity, the researchers propose the concept of location semantics. Researchers first divide the geographic space into a series of grid intervals, then count the query probabilities of users in different grid intervals from the history of location queries, use query probabilities as the quantified value of location semantics, select other locations with similar location semantics to the user’s location, and jointly construct the location anonymity set. These schemes are DLS [13], EDLS [13], MOS [19], K-DLCA [20], etc. For trajectory privacy protection, Wang et al. [21] construct a probabilistic model based on historical location data. The model calculates the location access probability and location transfer probability. For each benchmark location in the current user trajectory, k−1 locations with similar access and transfer probability are selected from the historical trajectory to construct the fake trajectory. Diao et al. [22] consider the temporal accessibility, historical query similarity, and in/out-degree of the trajectory based on the user’s historical trajectory to select a fake trajectory similar to the user’s trajectory. The above algorithms reduce the possibility of false locations or false trajectories being filtered out from the perspective of semantic similarity of locations.

Differential privacy [23] has gained attention in trajectory privacy protection due to its complete mathematical model and independence from the attacker’s background knowledge. Among them, in the statistical query service of spatial information, the use of spatial histograms [24] can decompose the geographic space into grid cells and record the location or trajectory segments of each grid cell. The use of a differential privacy algorithm can defend against attackers’ guessing attacks against location counts and trajectory counts of spatial cells by injecting noise into the histogram and publishing the data with the noise-added histogram.

In recent years, Geohash encoding technology has attracted the attention of researchers. It is a geolocation encoding technique proposed by Niemeyer [25]. It can convert a spatial two-dimensional geographic location (latitude, longitude) into a one-dimensional string identifier representing a spatial region. Location retrieval is achieved by one-dimensional string comparison while hiding the user’s precise geographic location in a location space region for location privacy protection. This technique is mainly applied to user location localization, nearest neighbor query, spatial object recognition, and location privacy protection. Wei [26] achieves location retrieval in the neighborhood POI problem by Geohash encoding the user’s spatial two-dimensional coordinates while hiding the user’s exact location. Liu et al. [27] used the Geohash code and pseudo-random sequence selection algorithm to select the geographical intervals involved in anonymous hiding from the location set and designed a k-anonymous location privacy protection scheme. Ye et al. [28] proposed a proximity detection method based on Geohash for privacy preservation. They first convert the user’s location coordinates into Geohash codes, which achieves a fast selection of candidate neighbors from a massive location set using Geohash codes with the same prefix, and then calculate the relative distance of candidate neighbors using the suffix. However, these methods only use Geohash codes instead of the user’s original location coordinates, without optimizing the retrieval process of the Geohash codes.

### 2.2. Spatial Indexing Techniques in Location-Based Applications

In spatial location-based applications and research, algorithms usually require efficient storage and frequent query and update operations for massive geolocation datasets, for which location-based spatial indices of objects are needed. There are object point access methods and object spatial access methods to establish spatial indices. The point access method generally uses the index structure of the binary tree, such as the KD tree [29], AK-D tree [30], KDB tree [31,32,33], LSD tree [34], etc., and its variants. Object space access methods convert the accessed objects into geometric mappings of points, lines, surfaces, and bodies representing their spatial location relationships for approximate indexing. Among them, object point mapping, object boundary qualification, and spatial segmentation are commonly used methods in object space access. Mortan codes and Hilbert codes [35,36] are often used in object point mapping methods, which use space-filling curves to map the locations of spatial objects to points in a one-dimensional space and encode them to create an index. The object boundary qualification method often uses the minimum bounding rectangle (MBR) of a spatial object to approximate the spatial relationship with an irregular shape, maps the objects into rectangular subspaces that define their spatial extent, and builds a spatial object index structure based on this. This method allows for the mutual overlap of subspaces due to the proximity, intersection, and overlap of objects in their spatial locations. These methods include R-tree [37,38,39] and its variants Hilbert R-tree [40], R*-tree [41], etc. Spatial partitioning methods generally use an iterative approach to grid partitioning of spatial extent, dividing large spaces into subspaces that do not overlap with each other and using step-by-step reduced subspaces to approximate object positions, such as B+ tree or quadtree.

A KD tree [29] is a balanced tree for searching a *k*-dimensional space, which is partitioned by iterative crossover until the *k*-dimensional space cannot be partitioned. Each iteration selects one of the dimensional elements, sorts them, and partitions the dimensional space into two subspaces using the median. Each node of the KD tree stores a location point in this *k*-dimensional space. The KD tree uses a trial-and-error put-back search algorithm to find the nearest neighbors of the target point and is now widely used in spatial range search and nearest neighbor search.

AK-D trees [30] improve runtime efficiency by using a depth-first search strategy for a non-tempt search for KD tree structures. KDB tree [31] is a modification of B-tree, a tree structure for *k*-dimensional search space, which combines the search efficiency of KD-tree and the block-oriented storage access capability of B-tree.

LSD tree [34] is a local split decision tree for *k*-dimensional space. Its structure is similar to the KD tree, but each node stores a local split decision composed of split dimension and position. LSD supports the efficient spatial localization of arbitrary spatial geometric objects and is suitable for spatial access path implementation in geometric databases.

R-tree [37] was proposed by Boston in 1984. It is a highly balanced tree in *k*-dimensional space and is now widely used in commercial spatial database systems. R-tree uses leaf nodes located at the bottom of the tree to store MBRs of qualified objects. Insert, query, and delete operations can be implemented simultaneously without index reconstruction.

B+ tree is an optimization of B-tree, which stores all data in leaf nodes only, and non-leaf nodes only store Key. It can keep data stable and ordered, and its insertion and retrieval have a log time complexity. The B+ tree has been widely used in databases and operating system file systems.

The quadtree [42,43] is a hierarchical spatial data structure. It is constructed based on the principle of spatial recursive decomposition. It splits a geographic region into four equal-sized sub-regions corresponding to four different quadrants: the southwest, northwest, southeast, and northeast. Each region is represented by a node in the tree. The quadtree is used widely in location retrieval, interest point recommendation, and location privacy protection.

### 2.3. Application of Spatial Indexing Techniques in Location Privacy Protection

In the study of location privacy protection, researchers have proposed various tree index structures to optimize the storage and retrieval of massive locations while hiding the user’s location. To et al. [44] addressed the existence of two phases of user retrieval and obfuscation processing in traditional location obfuscation techniques by designing a $B^{ob}$-tree index structure to simplify the process of location obfuscation algorithms and reduce the processing time. The $B^{ob}$-tree, designed based on the $B^{dual}$-tree, embeds geographically-aware area information on the tree nodes. The algorithm detects the obfuscated area containing the exact location of the user simply by traversing the $B^{ob}$-tree.

Zhang et al. [45] proposed the PrivTree method, which used an incomplete quadtree to partition the spatial locations and introduced an approximation error δ to reduce the noise error, which significantly improved the accuracy of the response experimentally on the Gowalla dataset.

Hu et al. [46] combined the spatial indexing capability of R-tree and the non-repudiation of Merkle Hash tree to design privacy-preserving authentication schemes for location range query services and improve the efficiency of location queries.

Zhao et al. [47] proposed a trajectory sequence structure tree SR-tree based on R-tree. Nodes in the tree store road sequences, location semantic features, user count statistics, location semantic count statistics, etc. SR-tree achieves fast retrieval of trajectories while preserving their spatiotemporal characteristics.

Yuan et al. [48] proposed a trajectory similarity tree structure based on the R-tree index structure to achieve spatial storage and query processing of trajectory data. They use differential privacy techniques to protect location privacy by adding noise to the user location statistics of the tree nodes.

Shao et al. [49] designed a Landmark Tree (LT) index structure for all location information to address the problem of sending the user’s exact location to anonymous servers in spatial range queries and proximity queries. The LT divides the map area into several landmarks and sends query requests with a combination of landmarks instead of the user’s exact location. This method achieves the purpose of hiding the actual user location within a specific query range.

For the query optimization problem of trajectory data, Gao et al. [50] designed the tree index structure of Geohash-Trees. They used Geohash encoding for adaptive grid partitioning of spatial regions according to the density of trajectories and encoded trajectories covering different grids to construct query trees and accelerate the efficiency of range queries.

Guo et al. [51] proposed the adaptive Hilbert-Geohash grid coding method AHG for the problems of spatial range query, nearest neighbor query, and object size query. AHG represents the location and approximate size of encoded objects by the grid hierarchy and the Geohash code length. The method achieves the efficiency improvement of spatial range query and neighbor query. Zhou et al. [52] address the location privacy protection in the nearest neighbor query by Geohash encoding the location of POI and constructing a trie tree structure at the server side, which takes the Geohash code of user location and Geohash code of POI location and performs character matching retrieval to achieve query optimization.

### 2.4. Comparative Analysis

In summary, researchers have proposed various tree index structures for different location privacy protection needs. The index structure using Quadtree, KD tree, R-tree, or their variants requires that the anonymous server can obtain the location coordinates of the requesting user. The anonymization scheme is constructed based on the assumption of the trustworthiness of the anonymous server, but this assumption is difficult to achieve in practical applications. The index structure using B-trees and B+-trees requires setting index keywords at each node in the tree and therefore requires a spatial dimension transformation for the location coordinates. Since the transformation process usually needs to be conducted on the anonymous server side, the users’ exact location still needs to be exposed to the server. Storing a massive number of locations, several of the above trees generally have a large depth, increasing the time consumption of retrieval. The rest, such as a landmark, semantics, prefix, and other trees, whose tree structure generally corresponds to the analysis process of geographic location semantics, can only be applied to specific occasions.

As a lightweight encoding algorithm, Geohash encoding can convert two-dimensional geographic coordinates into a one-dimensional string. It achieves the indexing of locations by string matching. The indexed result is a geographic region corresponding to a string, rather than an exact location coordinate. Thus, the exact location can be hidden in a geographic region. Different lengths of Geohash codes correspond to the different sizes of geographic regions. Therefore, by setting the length of the Geohash code, the purpose of personalized location region generalization can be achieved. The Geohash encoding process can be completed either at the user’s end or on the anonymous server. At the user’s end, the user’s side needs to convert the exact location to a string, then send a query to the server using the string as the requested location. In this way, the anonymous server cannot know the user’s exact location, and the need for the trustworthiness of the anonymous server is solved. However, this approach requires sacrificing some service accuracy. On the anonymous server, the user’s end sends the exact location coordinates to the server. This method assumes that the anonymous server is honest and trustworthy. However, the user can obtain more accurate query results. The GL-Tree tree combines Geohash encoding features with fewer layers of the tree limited by the maximum length of Geohash encode, thus reducing the time consumption of the retrieval process.

## 3. Geohash Encoding

### 3.1. The Idea of Geohash Encoding

Geohash is a geospatial coding technology with efficient retrieval capability, which is used widely in spatial distance retrieval. It can encode two-dimensional spatial coordinates (latitude and longitude) representing geographical locations into identifiers in the form of one-dimensional strings that can be sorted and compared. The idea is to first consider the Earth as a rectangular coordinate interval based on longitude (horizontal coordinate) and latitude (vertical coordinate), where the longitude ranges are (−180, 180) and the latitude ranges are (−90, 90), and then iteratively dichotomize and 0, 1 encode this rectangular coordinate region in the longitude and latitude directions.

### 3.2. Example of the Geohash Encoding Process

The following is an example of the Geohash encoding process. The geographic location is Beijing University of Posts and Telecommunications Building No. 1 in Haidian District, Beijing, with latitude 39.965391 and longitude 116.364017. The Geohash encoding process is described in Table 1.

Table 1 depicts the specific encoding procedure using location latitude coordinate as an example.

(1)The latitude interval (−90, 90) is iteratively dichotomized and area code, and after 20 rounds, the latitude value 39.965391 is finally encoded as a 20-bit binary string “1011100011010110101101111”, as shown in Table 1. Similarly, we may dichotomize and encode the longitude interval (−180, 180) iteratively, and after 20 rounds, we can encode the longitude value 116.364017 as a 20-bit binary string “1101001010101111110110”.(2)We cross-merge the latitude and longitude coded binary strings in (1) bit by bit in the order from left to right (the odd index bits of the merged binary string place the latitude coded bits, the even index bits place the longitude coded bits, and the index of the first position on the left is 0); then, we can obtain a combined 40-bit coded binary string is “1110011101001000110110111011111001111101”.(3)We split the combined 40-bit binary string into eight groups. Each group has five consecutive bits. Then, we convert each group into a decimal number. The result of the conversion is 28 29 4 13 23 15 19 29. We convert these decimal numbers into characters using base32 encoding. The final Geohash code is obtained, which is "wx4ergmx".

### 3.3. Geohash Encoding Algorithm Pseudocode

In this section, we give the pseudocode of the Geohash encoding algorithm as shown in Algorithm 1.
**Algorithm** **1:** Geohash Encoding Algorithm1: **Input:**
LR, range of geographic coordinates of the earth; Base32_Chars, an array of all characters encoded in BASE32; len, the length of the Geohash-code output by the algorithm; loc(lat,lng), the Location that need to be converted to Geohash-code, lat is latitude, lng is longitude.2: **Output:**
geohash_code, Geohash code of loc. 3: $binarystringlenth\leftarrow \left\lfloor \frac{len\times 5}{2}\right\rfloor$4: binarylat_length←binarystringlenth5: binarylng_length←binarystringlenth6: **if**
len is an odd number **do**:7:    binarylng_length←binarystringlenth+18: **end if**9: binarylat←Φ,binarylng←Φ10: latmin←−90,latmax←90,lngmin←−180,lngmax←18011: currentlen←012: **while**
currentlen<binarylat_length**do**:13:     $middle\leftarrow \frac{latmin+latmax}{2}$14:     **if** $lat\in \left[latmin,middle\right)$ **do**:15:          $binarylat\leftarrow binarylat\cup \left\{‘0’\right\}$16:          latmax←middle17:     **else**18:          $binarylat\leftarrow binarylat\cup \left\{‘1’\right\}$19:          latmin←middle20:     **end if**21:     currentlen++22: **end while**23: currentlen←024: **while**
currentlen<binarylng_length**do**:25:     $middle\leftarrow \frac{lngmin+lngmax}{2}$26:     **if** $lng\in \left[lngmin,middle\right)$ **do**:27:          $binarylng\leftarrow binarylng\cup \left\{‘0’\right\}$28:          lngmax←middle29:     **else**30:          $binarylng\leftarrow binarylng\cup \left\{‘1’\right\}$31:          lngmin←middle32:     **end if**33:     currentlen++34: **end while**35: binarystringlenth←len×536: binarylatandlng←Φ37: **for**
i←1tobinarystringlenth**do**:38:     **if** *i* is an odd number **do**:39:        binarylatandlng←binarylatandlng∪getABitByOrder(binarylng)40:     **else**41:       binarylatandlng←binarylatandlng∪getABitByOrder(binarylat)42:     **end if**43: **end for**44: String[]substrings←splitBinaryString(binarylatandlng,5)45: int[]indexs←getIndexs(substrings)46: geohash_code←ϕ47: **for**
i←0toindexs.length−1**do**:48:     geohash_code←geohash_code∪findCharFromBase32(indexs[i])49: **end for**50: **return** 
geohash_code

The execution process of Algorithm 1 is described as follows.

(1)In lines 3–8, the algorithm initializes the length of the binary strings of latitude and longitude according to len.(2)In lines 9–34, the algorithm defines two empty sets for recording, respectively, the binary character set obtained from the location latitude and longitude, according to the Geohash encoding conversion method.(3)In lines 35–43, the algorithm cross-merges the latitude and longitude binary strings bit by bit in the order from left to right. When merging, a character ‘1’ or ‘0’ is taken from the longitude binary string in order when *i* is an odd number, and a character ‘1’ or ‘0’ is taken from the latitude binary string in order when *i* is an even number or ‘0’.(3)Lines 44–49, in left-to-right order, the algorithm splits the merged binary string by one segment every five bits. Then, the segmented binary string is converted into a decimal integer, and the decimal integer is used as an index value to find its corresponding ASCII character from base32. Then, the algorithm generates the Geohash code.

### 3.4. Analysis of the Geohash Encoding Process

Based on the above process of Geohash encoding, we can analyze the following characteristics of Geohash encoding:(1)Geohash encoding is performed by iteratively dichotomizing the latitude and longitude intervals on the Earth’s surface, gradually narrowing down the intervals to keep approximating a geographic coordinate position. Therefore, a specific two-dimensional spatial location coordinate can be encoded into a unique Geohash-encoded string. However, a Geohash-encoded string is a representation of a rectangular location region, not a specific location point. Different encoding lengths correspond to different sizes of segmentation areas, and the shorter the encoding length, the larger the area represented. Table 2 shows the variation of latitude, longitude, and geographical extent of the spatial rectangular interval represented by the code as the length of the Geohash code increases.

(2)If two different Geohash codes have the same string prefix, it means that the geographical interval corresponding to these two Geohash codes is two different subintervals of the interval corresponding to their common prefix string Geohash code. For example, the following nine Geohash codes: “wx4ergmq”, “wx4ergmw”, “wx4ergmy”, “wx4ergmr”, “wx4ergmx”, “wx4ergmz”, “wx4ergt2”, “wx4ergt8”, and “wx4ergtb”, which have six bits of the same string prefix “wx4erg”, indicate that they belong to the Geohash subintervals of the region encoded as “wx4erg”.(3)As the length of the Geohash code increases by 1 for each geographic interval corresponding to the encoding, the geographic interval will be divided into 32 grid intervals of the same size. The grid is divided into 4×8 for odd bits and 8×4 for even bits, as shown in Figure 1a,b.(4)The coded characters are space-filled according to the “Z” curve. Neighboring characters correspond to the spatial location of the grid nearby. However, there is a case of abrupt change in position. As shown in Figure 1, the encoded characters “7” and “8”, “g” and “h”, “s” and “r”, although the characters are adjacent to each other, the spatial positions are far away.

## 4. GL-Tree

### 4.1. Structure of GL-Tree

To achieve efficient storage and retrieval of massive amounts of location data, we use Geohash encoding to convert the two-dimensional location coordinate into a one-dimensional Geohash code string and design an *L*-level tree data storage structure (named GL-Tree). The GL-Tree sets the value of the tree level *L* at initialization. The value of *L* is equal to the maximum length of the Geohash code string of the location. The structure of the GL-Tree is shown in Figure 2. It has the following characteristics.

(1)The GL-Tree is composed of many L-Trees. The L-Tree is a four-layer structured tree, the structure of which is shown in the double-dotted rectangular box in Figure 2. The L-Tree is labeled as L-level according to the level located in the GL-Tree. If an L-Tree is considered a node, the GL-Tree is a multi-branch tree structure composed of these nodes.(2)Each L-Tree consists of four layers of structure, from the top down, the root node, the first-layer middle node, the second-layer middle node, and the leaf node.(3)Each L-Tree has only one root node. The root node consists of a four-tuple $\left(l_{i},pre,mp,pt\right)$. The $l_{i}$ is the level of the L-Tree in the GL-Tree. The pre is the prefix string of the Geohash code corresponding to this L-Tree. The mp is a pointer to the middle node at the top layer of the L-Tree. The pt is a pointer to the root node of the previous L-Tree in the GL-Tree.(4)The middle node of the L-Tree is divided into two layers. Each node sets the keywords for branch retrieval. Each keyword corresponds to one branch. The number of keywords in the first layer of nodes is no more than four, and the number of keywords in the second layer of nodes is no more than two. A keyword is a character in Base32 encoding. That is, each node has at most four or two keywords represented by Base32 characters. These characters are arranged left to right on the middle node in the order they are in the Base32 encoding.(5)A branch corresponding to one keyword is connected by a pointer*p*. Furthermore, *p* points to the next middle node or leaf node in the L-Tree.(6)In the middle node of the L-Tree, the keyword is the largest character of the next node pointed by the pointer *p*.(7)The leaf nodes of the L-Tree are used to store the list of geographic locations. All locations can only be stored in the leaf nodes of the lowest level L-Tree of the GL-Tree, or in the leaf nodes of the L-Tree at any level, according to the actual application requirements and user location privacy needs.(8)Each leaf node consists of at most four data items and two pointers pd and nd. All the leaf nodes in each L-Tree are connected by pd and nd into a bidirectional link sorted in keyword order. A head node is set as the first node of this link.(9)Each data item of the L-Tree is a three-tuple consisting of a keyword, a list of geographical locations ul, and a pointer nt to the next level of the L-Tree in GL-Tree.(10)The geographic location is stored in ul, and a geographic location is stored only once in the same data item.

### 4.2. The Physical Meaning of the GL-Tree

According to the Geohash encoding analysis, as the length of the Geohash code increases, its corresponding geospatial area gradually shrinks. In a physical sense, extending the GL-Tree tree from top to bottom is equivalent to hierarchical gridding of the geographic location space. The following is an explanation of this.

(1)As the hierarchy of GL-Tree extends deeper, the smaller the grid interval it divides the geographic location space into, the larger the length of the Geohash code corresponding to the grid interval. The L-Tree structure of the *i*th level of the GL-Tree tree corresponds to different choices of the *i*th character bit of the Geohash code. According to the base32 encoding and Geohash encoding rules, there are 32 possible values for each character bit of the Geohash code. Therefore, at each level of the GL-Tree, the keywords in the data items of each L-Tree also have 32 possible character fetching values. Since each data item is connected to a next-level L-Tree by a pointer nt, there are at most $32^{i-1}$ L-Trees at level *i* of the GL-Tree.(2)Each L-Tree in a GL-Tree corresponds to a geographic location interval. For example, if the pre value in the root node of an L-Tree is “wx4er”, this means that the L-Tree is located at the 6th level of the GL-Tree tree and corresponds to a rectangular Geohash code of “wx4er” geographic location interval. At this level, the location interval will be further gridded into 32 same-size subintervals.(3)A data item in an L-Tree corresponds to a Geohash code, which also corresponds to a rectangular geolocation interval. This interval is one of the 32 subintervals of the same size obtained by further gridding the rectangular interval corresponding to the L-Tree. The length of the Geohash encoding corresponding to the data item of the L-Tree at level *i* of the GL-Tree tree is also *i*. For example, if the example L-Tree in (2) has a data item with the keyword “7”, it corresponds to the subinterval of the Geohash code “wx4er7”.(4)The deeper the L-Tree is in the GL-Tree, the smaller the rectangular geographic interval corresponding to the data items of that L-Tree is. The geographic interval corresponding to the data items of the upper L-Tree is further divided into 32 smaller subintervals in the lower L-Tree pointed by the pointer tp.(5)The L-Tree structure is designed based on the characteristics of the Geohash encoded division of location intervals in spatial proximity relationships. As can be seen in Figure 1, the coded corresponding intervals have the following characteristics: adjacent numbered grid intervals are also adjacent positions; for every four adjacent numbered intervals in a group, the positions of the adjacent numbered intervals behind will have smaller mutations; for every eight adjacent numbered intervals in a group, the positions of the adjacent numbered intervals behind will have larger mutations. In order to make the L-Tree reflect this inter-interval position relationship, we design the structure of the L-Tree as a “4-2-4” structure, as shown in Figure 2. The first layer intermediate node contains at most four branch terms corresponding to 32 consecutive numbers, which can manage the division and position storage of the whole interval corresponding to the L-Tree. The second layer intermediate node contains at most two branching items, corresponding to eight consecutive numbers, which can manage the division and location storage of eight adjacent intervals. The leaf node has four data items, corresponding to four consecutive numbers, which can manage the division and location storage of four adjacent intervals.(6)In an L-Tree, if the nt of a data item is not null, it points to an L-Tree in the next level of the GL-Tree, indicating that the geographic interval corresponding to the data item is further divided into smaller subintervals.(7)The pre is in the root node of the lower L-Tree, whose value is equal to the Geohash code of the geographic interval represented by the upper L-Tree data item pointing to this root node.(8)For the L-Tree at level *i*th of the GL-Tree, the process of traversing from the root node to a data item of the L-Tree is the process of retrieving and matching the position of the *i*th character in a Geohash code.(9)The Geohash code of the geographic interval corresponds to the data item in the L-Tree. The Geohash code corresponds to a retrieval route. This retrieval route starts from the root node of the L-Tree at the first level of the GL-Tree and goes down until the data item is retrieved. In retrieval order, the keywords of all the data items passing through the retrieval route are connected by a string. The string is the Geohash code of the geographic interval corresponding to the data items.(10)The character matching process for the intermediate nodes of the L-Tree should be performed along the first keyword to its left, which is not smaller than the retrieved character. For example, the four keywords of the first intermediate node in Figure 2 are “7”, “g”, “r”, and “z”. When searching for the character in a Geohash code, if the corresponding character in the Geohash code is any character from “1” to “7”, the search is performed down the pointer *p* of the keyword “7”. Furthermore, if the character is any character from “8” to “g”, the search is performed down the pointer *p* of the keyword “g”. The retrieval process of “r” and “z” is similar.(11)The storage of geographic location in GL-Tree is to perform a keyword search on the GL-Tree tree according to the Geohash code of the location coordinates, in top–down order, to find the L-Tree corresponding to the Geohash code, and to find the corresponding data item in this L-Tree. Then, the location is stored in the ul of this data item. The location search then follows the same process to find the location stored in the ul of the corresponding data item.

### 4.3. Example of Retrieval and Maintenance Process of GL-Tree

To improve retrieval efficiency and reduce storage space, GL-Tree is constructed, stored, and retrieved dynamically. The stored procedure is to find the location of the corresponding store data item in the GL-Tree. The following is an example to illustrate the location retrieval process.

Let us make two assumptions. First, the maximum level *L* of the GL-Tree tree is set to 8 during initialization, which indicates that the length of the Geohash code is also 8. Furthermore, the code corresponds to the minimum geographic interval obtained after gridding the geographic space. Second, the retrieved geographic coordinates are loc = (latitude: 40.222012, longitude: 116.248283), and the location coordinates have been stored in GL-Tree. The following is the specific location retrieval process.

(1)The Geohash encoding algorithm is first executed to convert the geographic coordinates (latitude: 40.222012, longitude: 116.248283) into the corresponding 8-char Geohash code “wx4sv61q”. According to the Geohash encoding precision, the string code corresponds to a rectangular geolocation interval of size 4.9×4.9 km^2^.(2)For location retrieval, the retrieval algorithm reads each character from the Geohash code in turn, and then searches the L-Tree at the corresponding level in the GL-Tree to find the data item corresponding to each character. The algorithm firstly reads the first character “w” of the Geohash code, then starts from the root node of the L-Tree at level 1 in the GL-Tree, searches along the tree structure, top–down, and finds the data item whose keyword is “w” in the leaf nodes of the L-Tree. If the pointer nt of this data item is not null, the algorithm finds the root node of the second-level L-Tree along this pointer and continues to retrieve the data item corresponding to the character “x” in the second level L-Tree. In this recursion, the data items corresponding to characters “4”, “s”, “v”, “6”, “1”, and “q” are retrieved in the L-Tree at levels 3–8 in turn.(3)If the algorithm finds the data item corresponding to the character "q" in the 8th level L-Tree, it can find the coordinates of that location from the ul of the data item. In the case of location storage, the location coordinates can be stored in this ul.(4)To improve the retrieval efficiency of the algorithm and reduce the consumption of memory space, the GL-Tree tree is dynamically constructed and maintained. If the data item corresponding to the keyword is not found in any layer of the L-Tree, or if nt in the data item is empty, this implies that this data item and its subsequent L-Tree do not exist, and the current location to be retrieved is not stored in the GL-Tree. For example, a. If the data item corresponding to the keyword “v” is not found in the fifth level L-Tree, this indicates that the L-Tree corresponding to the Geohash code “wx4s” exists in the GL-Tree, the L-Tree corresponding to the Geohash code “wx4sv” and its subsequent L-Trees do not exist. b. If the data item corresponding to the keyword “v” is found in the fifth level L-Tree, but the nt of this data item is null, that means the data item with the Geohash code “wx4sv” exists in the GL-Tree at this time and there is no subsequent L-Tree. Therefore, if the algorithm is performing a location retrieval operation and returns null, this indicates that it failed to retrieve the location coordinates. If the algorithm is performing a location storage operation, case a requires inserting a new data item with the keyword “v” at the fifth level L-Tree and creating structure and data items of the L-Trees at its subsequent levels, while case b requires recursively creating structure and data items of the L-Trees at layers 6–8 below the data item with the keyword “v”. In the end, the position is stored in ul of the data item with the keyword “q” in the 8th level L-Tree.(5)The process of searching at the intermediate nodes of the L-Tree has been described in (9) of the previous section. The description will not be repeated here.

## 5. Algorithm and Analysis

In this section, we describe the construction algorithm and location retrieval algorithm of GL-Tree using pseudocode and analyze the performance of the algorithms.

### 5.1. Construction Algorithm for GL-Tree

The GL-Tree construction algorithm constructs a GL-Tree in memory based on the location dataset and stores the locations into the corresponding data items. The algorithm is implemented by several sub-algorithms recursively calling and combining each other. The essence of the algorithm is to initialize an empty GL-Tree with only the first level L-Tree root node. Then, the locations from the dataset are inserted into this GL-Tree. During the process of location insertion, the creation of the L-Trees and their root nodes, middle nodes, leaf nodes, and data items at each level is completed, and the locations are added to the ul list of the corresponding data items. The pseudocode is shown in Algorithm 2 and its sub-algorithms below.
**Algorithm** **2:** Construction Algorithm for GL-Tree1: **Input:** LS, a location dataset; level, the maximum hierarchical value of the GL-Tree.
2: **Output:**  gltree, a GL-Tree structure which has stored the locations in the location dataset to the corresponding data items. 3:  /* Initialize an empty GL-Tree with only the topmost L-Tree root node, without any location data */4:  gltree←initGLTree(level)5:  /* Read all the locations in the location dataset and store them in the GL-Tree */6:  **for**
i←0toLS.size()−1**do**:7:       loc←Ls.get(i)8:       gltree.insertLoc(loc)       // Insert a location into the GL-Tree, this method will call Algorithm 3 within9:  **end for**10:  **return**
gltree

In Algorithm 2, line 4, the algorithm initializes an empty GL-Tree based on setting the maximum level value level. It has only the topmost L-Tree root node and does not store any location data. In lines 6–9, each position in the position dataset is read cyclically and added to the GL-Tree. Line 8 calls the method insertLoc of the GL-Tree class to insert a location into the GL-Tree. Furthermore, inside of the method further calls the insertLoc method of the L-Tree class to insert the location into the L-Tree. Algorithm 3 describes the pseudocode of the insertLoc method in the L-Tree class.

The execution process of Algorithm 3 is described as follows:(1)Line 3, to convert loc to its corresponding Geohash code.(2)Line 4, to read the character corresponding to the position in the current hierarchy from the Geohash code.(3)Line 5, to find the leaf node corresponding to character *c* in the current L-Tree.(4)Lines 6–17, if the level of the current L-Tree is the lowest level of the GL-Tree, the algorithm determines the presence or absence of the data item corresponding to *c*. If yes, then insert the location loc into the ul list of that data item. Otherwise, a new data item is created in that L-Tree using the keyword *c*, and the location loc is added to the ul list of the data item.(5)Lines 18–34, if the current level of the L-Tree is not the lowest level of the GL-Tree, the algorithm determines the presence or absence of the data item corresponding to *c*. If it exists, the algorithm inserts the location loc into the ul list of that data item, and the location insertion operation continues at the next level of the L-Tree pointed by nt in the data item. Otherwise, in that L-Tree, create a new data item using the keyword *c* and insert the position loc into the ul list of the data item, then continue to create a new next-level L-Tree and insert the location loc in the next-level L-Tree.(6)As a side note, the algorithm described here stores the locations in the leaf nodes of the L-Tree at each level. The purpose is to verify the performance of storing and querying locations in different levels of the L-Tree. However, for specific application scenarios, the location loc can be inserted into the ul list of the L-Tree at the specified level only according to the actual requirements.
**Algorithm** **3:**  Insert a location into L-Tree1: **Input:** loc, a location in location dataset; currentlevel, the hierarchical value of the current L-Tree in the GL-Tree for which the algorithm is called; level, the maximum hierarchical value of the GL-Tree.2: **Output:**  3:  geohash←getGeohash(loc)4:  c←geohash.charAt(currentlevel−1)5:  leafnode←findLeafNodeFromLTree(c)6:  **if** currentlevel is equal to level **do**:7:      **if** leafnode not is null **do**:8:           dataitem←leafnode.findDataItemFromLeafNode(c)9:           **if** dataitem not is null **do**:10:         dataitem.addLoc(loc)11:         **else**12:            leafnod.addNewDataItem1(c,loc)13:         **end if**14:      **else**15:            addNewLeafNode1(loc)16:            modifyLeafNodeList()17:      **end if**18:  **else**19:      **if** leafnode not is null **do**:20:         dataitem←leafnode.findDataItemFromLeafNode(c)21:         **if** dataitem not is null **do**:22:            dataitem.addLoc(loc)23:            nextltree←dataitem.getnextLTree()24:            nextltree.addLoc(loc,currentlevle+1,level)25:         **else**26:            leafnod.addNewDataItem2(c,loc)27:            newtltree.addLoc(loc,currentlevle+1,level)28:         **end if**29:      **else**30:         addNewLeafNode2(loc)31:         modifyLeafNodeList()32:         newtltree.addLoc(loc,currentlevle+1,level)33:      **end if**34:  **end if**35:  **return**

### 5.2. Location Retrieval Algorithm

In various location-based application services and location privacy protection schemes, a core function is to retrieve a specified location or location subsets from a massive number of geographic locations. The retrieval algorithm performs a fast retrieval for a specified location stored in the GL-Tree. The algorithm uses the Geohash code of the location to find the geographic interval in which the location is located. It avoids the matching process for a large number of irrelevant locations and improves the efficiency of location retrieval by compressing the size of the location interval and narrowing the retrieval range. Algorithm 4 describes the pseudo-code of the location retrieval algorithm.
**Algorithm** **4:** Location retrieval algorithm1: **Input:** loc, a target location; searchlevel, possible hierarchy of target locations in the GL-Tree; gltree, a GL-Tree structure which has stored the locations in the location set to the corresponding data items.2: **Output:** flag, a boolean variable indicating whether the target location was retrieved or not 3:  geohash←getGeohash(loc)4:  currentLTree←gltree.topltree5:  currentlevel←16:  **While** currentLTree not is null and currentlevel<=searchlevel **do**:7:      c←geohash.charAt(currentlevel−1)8:     leafnode←currentLTree.findLeafNodeFromLTree(c)9:      **if** leafnode not is null **do**:10:         dataitem←leafnode.findDataItemFromLeafNode(c)11:         **if** dataitem not is null **do**:12:            **if** currentlevel is equal to searchlevel **do**:13:               flag←dataitem.findLocIndataitem(c)14:            **else**15:               currentLTree←dataitem.getnextLTree()16:               currentlevel++17:            **end if**18:         **end if**19:     **end if**20:  **end while**21:  **return**
flag

The execution process of the Algorithm 4 is described as follows:(1)Line 3, first converts target loc to its corresponding Geohash code.(2)Line 4, locates the top-level L-Tree of the GL-Tree and starts the retrieval operation from this tree.(3)Line 5, sets the hierarchy of the current query L-Tree in the GL-Tree.(4)Lines 6–20, If the currently retrieved L-Tree is not null, the algorithm extracts the character *c* at the position corresponding to the Geohash code of loc and then verifies whether the leaf node corresponding to *c* exists. If the leaf node exists, the algorithm continues to find whether there is a data item with the keyword *c* in the leaf node. If the data item with the keyword *c* exists, this case indicates the current level is the target level. Then, it continues to retrieve whether there is a target loc in the list ul of the data item. Therefore, it returns true or false. If the level of the retrieved L-Tree is smaller than the target level, the algorithm moves to the next level of the L-Tree along the pointer nt in the data item with keyword *c* and continues retrieving. If the leaf node or data item corresponding to *c* in the retrieved L-Tree does not exist, it returns false directly.

### 5.3. Algorithm Analysis

After the GL-Tree is created, the algorithm reads the location data from the location dataset, calculates the Geohash code of the location, dynamically creates the L-Tree and its related nodes at the corresponding level based on each character in the Geohash code, and stores the location in the data item with that character as the keyword. In the extreme case, the GL-Tree is a complete tree, i.e., the *i*th level consists of $32^{i-1}$ complete L-Trees. Each complete L-Tree contains one root node, five middle nodes, eight data nodes, and thirty-two data items. Therefore, a complete GL-Tree with *n* levels has $\sum_{i=1}^{n}32^{i}$ data items, $\sum_{i=1}^{n}8^{i}$ data nodes, and $\sum_{i=1}^{n}5^{i}$ middle nodes. The storage overhead is an exponential complexity concerning the level *n*, denoted as $O(a^{n})$. It needs a significant amount of memory space to store the GL-Tree data structure. The population distribution on the earth’s surface is extremely uneven because a large number of areas are sparsely populated, such as oceans, mountains, swamps, forests, deserts, etc. In the practical application of location-based services, the geographical area that can be covered by the location dataset collected by the application is much smaller than the surface of the Earth. The GL-Tree dynamically constructed from such a location dataset is an incomplete tree. It has a more simple structure than a complete tree, and the number of data items is much smaller than the $\sum_{i=1}^{n}32^{i}$.

### 5.4. Analysis of Algorithm Performance

The location retrieval algorithm starts from the root node of the first level of the L-Tree in the GL-Tree to perform character lookups along the GL-Tree level-by-level from the top down, matching keywords in each corresponding level L-Tree to find data items. In the worst case, the times of keyword match in each intermediate node of the L-Tree is six, and the times of keyword match in each data node is four, for a total of ten times. To find the location data stored in the GL-Tree with *n* levels, the times of keyword match is 10×n. Therefore, the time complexity of location retrieval is O(n). It is independent of the total number of locations stored in the GL-Tree.

The GL-Tree construction algorithm requires reading locations from the location dataset, inserting them into the data items of the corresponding L-Tree in the GL-Tree, and dynamically creating nodes when the nodes do not exist. The time complexity of the algorithm is $O(ma^{n}+n)$ in the worst case, using a location dataset containing *m* locations to construct a complete GL-Tree with *n* levels. Since *n* is much smaller than *m* in general, the time complexity can be simplified to $O(ma^{n})$.

### 5.5. Analysis of Algorithm Application

GL-Tree is applied to store a large number of locations or retrieve specified locations from them. They are stored in data items corresponding to rectangular Geohash encoded intervals. The key to fast retrieval is to retrieve location intervals rather than directly retrieve specific location coordinates. Since the Geohash encoding divides the location interval, there is a problem of mutation of adjacent encoded intervals in spatial proximity. Therefore, there is a need for location retrieval to cross data items, leaf nodes, and L-Trees. When location retrieval is required to cross data items and leaf nodes, the algorithm can rapidly locate other data items and leaf nodes along a bi-directional chain table composed of all data items of the L-Tree after finding a data item in the target interval. When location retrieval is required to cross the L-Tree, the pt pointer in the root node of the L-Tree can be used to expand the search scope to its upper-level L-Tree, which corresponds to a larger location interval.

Based on the above analysis, it can be seen that when developing location-based object retrieval applications for different needs, the retrieved objects, and their locations, can be stored in the data items of GL-Tree, and the object retrieval service can be achieved by compressing and retrieving the location intervals where the objects are located. For example, point-of-interest query, nearest neighbor query, object range query, location-based recommendation system, etc.

Using location intervals to hide the exact location is a common approach in location privacy protection schemes, since the data items of different levels L-Tree in GL-Tree correspond to different sizes of Geohash encoding intervals. Therefore, an anonymous server can be used to create the GL-Tree in designing a location privacy protection scheme. After receiving a user’s query request and personalized privacy requirements, the anonymous server extracts the user’s accurate location from the query request and inserts the user’s query and location into data items of different levels of the L-Tree according to the size of the hidden interval set by the user. Then, the anonymous server sends a query to the location server, using the Geohash code of the interval in which the user is located instead of the user’s specific location. The purpose of location privacy protection can be achieved in this way.

## 6. Experiments and Results Analysis

This section tests the performance of GL-Tree in tree construction, location insertion, and location retrieval through experiments. The experiments use the Gowalla public dataset. The dataset is a US Twitter friend relationship dataset and a location check-in dataset. We extract location coordinates from the location check-in dataset and perform de-duplication. Then, 1,250,000 location coordinates are randomly selected from the processed results to generate the experimental dataset. Figure 3 shows that the geographical location distribution of the experimental dataset is remarkably uneven. Therefore, the created GL-Tree using this dataset is an incomplete tree. We conducted comparison experiments for the B+ tree, R tree, and GL-Tree to evaluate and compare their time consumption in tree construction, location insertion, and location retrieval. All experiments were repeated 100 times. The test results were averaged.

We used java language in the eclipse development environment to complete the code writing and experimental testing. The experiment is running on Ali Cloud Elastic Compute Server (ECS), with 4 cores (vCPU) Intel(R) Xeo(R) Platinum 8369B CPU @2.70 GHz, 16 GiB memory, 64-bit windows server 2019 datacenter OS, and JavaSE-1.8 environment.

### 6.1. Comparison of the Impact of the Amount of Location Data

The experiments are constructed dynamically for both B+ trees, R-trees, and GL-Trees. We perform incremental partitioning of the dataset for the experimental dataset in increments of 50,000 segments. The time consumption of the B+ tree, R tree, and GL-Tree in tree creation, location insertion, and location retrieval are tested for comparison when the total number of locations in the dataset is incremented. The experimental results are shown in Figure 4.

In Figure 4, the time consumption of all three tree structures shows an increasing trend as the number of locations increases. The time consumption of the R tree grows fastest, while it grows relatively slowly in the B+ Tree and GL-Tree. For the same amount of location data, RTree has the highest time consumption and B+ tree has the lowest time consumption. The time consumption of GL-Tree is about the average of RTree and B+ tree. In position insertion and position query, the time consumption of the R tree is still much larger than the other two trees, while GL-Tree is closer to the B+ tree, but GL-Tree is comparatively lower. In terms of stability, when the number of locations exceeds 350,000, the time consumption of the three trees fluctuates around the average value for both the position insertion operation and the position query operation. There is no significant increasing trend with the increase in the position amount, but the B+ tree and GL-Tree are more stable. The average values of the experimental results for inserting location data are 17.33 μs, 4.08 μs, and 2.42 μs, respectively, while the average values of the experimental results for location query are 4.82 μs, 3.60 μs, and 3.28 μs, respectively.

### 6.2. The Effect of Location Order on the Performance of GL-Tree in Location Dataset

The algorithm reads the locations in the dataset sequentially to construct the structure of the GL-Tree. Changes in the location reading order affect the order of the nodes created during the construction of the tree. We use all 125,000 location data to test the performance of GL-Tree by changing the location order in the dataset. For each experiment, we randomly disrupt the order of all the locations in the dataset before creating the GL-Tree. Without loss of generality, after the creation of the tree is finished, we perform a position retrieval operation by randomly selecting a position from the dataset. A new location not in the dataset was selected for the location insertion operation. The experiment was performed 200 times and the results are shown in Figure 5.

As can be seen from the figure, the time consumption for GL-Tree creation, location insertion, and retrieval fluctuates but exhibits a smooth overall trend when the location order in the location dataset is changed arbitrarily. The average time consumption of GL-Tree for tree creation, location insertion, and retrieval are 4.75 s, 3.62 μs, and 2.64 μs, respectively.

### 6.3. Analysis of Experimental Results

The GL-Tree uses the character encoding by Geohash as the retrieval keywords of the tree nodes. Each hierarchical L-Tree in the GL-Tree corresponds to the retrieval process of one character of the Geohash code. According to this rule, the depth of the GL-Tree is limited, and the depth of the hierarchy is equal to the length of the Geohash code. At the same time, in GL-Tree, the L-Tree of one level corresponds to a location interval, and the data item of this L-Tree corresponds to a small sub-interval of $\frac{1}{32}$ of this location interval. GL-Tree stores all the locations within this small sub-interval in a unified geolocation list ul of the same data item. ul’s structure can be designed as an ordered list or a tree structure, which depends on the number of locations stored in the location sub-interval in a specific application to simplify storage or optimize queries.

The B+ tree needs to use an index that can be sorted during the construction and retrieval. In our experiments, we use the Geohash code of the location as the index and store the location of the same Geohash code on the same leaf node of the B+ tree. Therefore, the leaf nodes at the bottom level of the B+ tree correspond to the physical meaning of the leaf nodes at the lowest level of the L-Tree in the GL-Tree.

All the tree structures in the experiments are created dynamically, i.e., the corresponding hierarchical nodes are generated dynamically according to the demand of the stored locations. Therefore, for the B+ tree and GL-Tree, when the stored locations reach a certain number, the nodes in the tree already contain all the intervals of the locations in the dataset. At this point, when continuing to insert new locations, the number of nodes and the depth of the tree no longer increases, and the structure of the tree no longer changes, so the time consumption of location insertion and retrieval operations will converge and will not grow with the increase in the number of locations.

R-tree is mainly for the storage and retrieval of objects with certain location spaces, the nodes correspond to the minimum bound rectangle (MBR) rectangular location interval containing their stored objects, and the space area corresponding to the sibling nodes of R-tree can overlap; this overlap causes the space index potentially having to search for multiple paths before the final result is obtained, thus greatly reducing the efficiency, and in the worst case first, its time complexity can degrade from logarithmic search to linear search.

In addition, the leaf nodes of the L-Tree at each level in the GL-Tree can be employed for location storage. Therefore, according to the location privacy protection requirement or the size of the geographic location interval covered by spatial objects, differentiated location storage and retrieval needs can be provided by storing locations in the leaf nodes of different levels of L-Tree. This feature is not available in either B+ trees or R trees. In addition, due to the use of location intervals, GL-Tree enables unified batch management and retrieval of location sets that are in the same location interval.

## 7. Conclusions

The storage, management, and retrieval of massive geographic locations have a wide range of needs in the application of GIS, location-based service applications, and location privacy protection. To achieve efficient storage and retrieval for locations, neighboring location sets, and geographical range of spatial objects, and to meet the need for differentiated location privacy protection, we design a hierarchical tree structure, GL-Tree. The GL-Tree utilizes the features of Geohash codes to achieve storage, retrieval, and hiding of locations by managing and retrieving location intervals corresponding to Geohash codes, and unified batch management and retrieval of location sets in the same location interval. In addition, it is also possible to hide locations in Geohash code location intervals with different space sizes to achieve the need for personalized location privacy protection. We have further elucidated the performance of GL-Tree in tree creation, location insertion, and location retrieval through comparative experiments and analysis.

In this paper, we focus on the structure and performance of GL-Tree and its role in location privacy protection. Future work will consider the following points to improve the GL-Tree tree and enhance its usability and security performance.

The future primary work is to design a GL-Tree location privacy protection scheme based on the nearest neighbor query and spatial range query as application scenarios and perform performance tests.Improve GL-Tree to make it easy to record users’ trajectories and location semantics, and achieve efficient retrieval and similarity comparison of trajectories.Combine the non-repudiation of Merkle tree with the storage retrieval performance of GL-Tree. Add service authentication in location privacy protection using blockchain technology. Implement a multi-chain location privacy protection scheme using public and federated chains to decentralize user queries and location storage based on user location privacy levels.To combine GL-Tree and differential privacy techniques to design a location privacy protection scheme based on differential privacy. According to the user’s location sensitivity requirement, *k* intervals with similar access frequency to the user’s location interval are selected, and Laplace noise is added to the location data of the corresponding nodes.

## Figures and Tables

**Figure 1 sensors-23-02245-f001:**
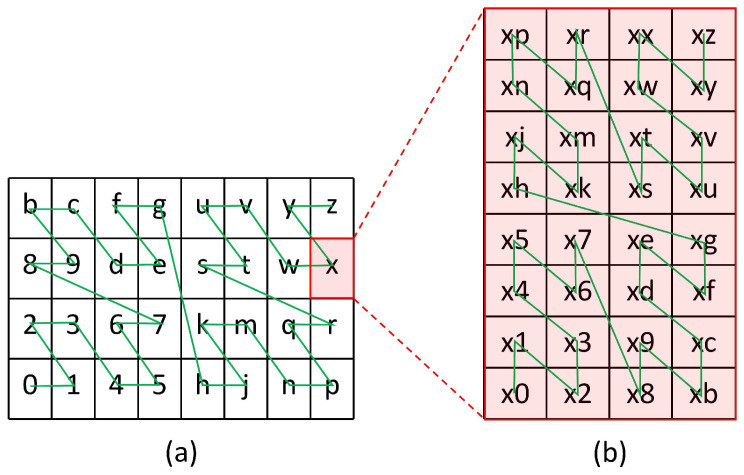
The grid division method corresponding to the coding position. (**a**) When Geohash is encode with odd bits, the spatial region division method. (**b**) When Geohash is encode with even bits, the spatial region division method.

**Figure 2 sensors-23-02245-f002:**
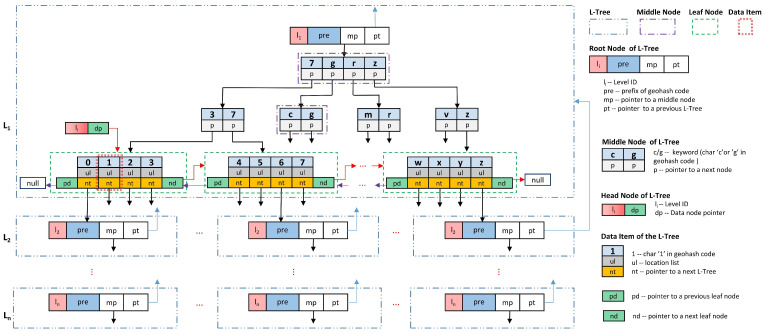
The structure of GL-Tree.

**Figure 3 sensors-23-02245-f003:**
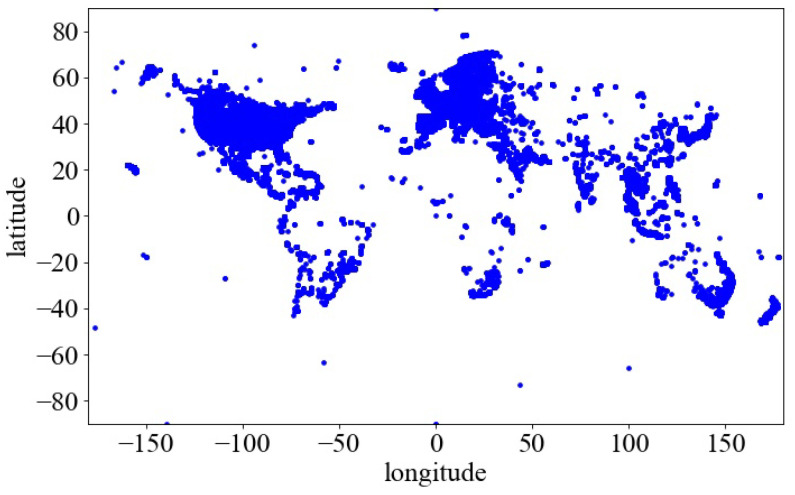
Geospatial distribution map of the locations.

**Figure 4 sensors-23-02245-f004:**
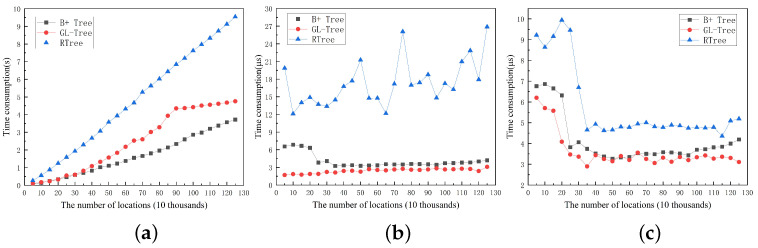
The impact of the amount of location data. (**a**) Creation GL-Tree. (**b**) Location insertion. (**c**) Location retrieval.

**Figure 5 sensors-23-02245-f005:**
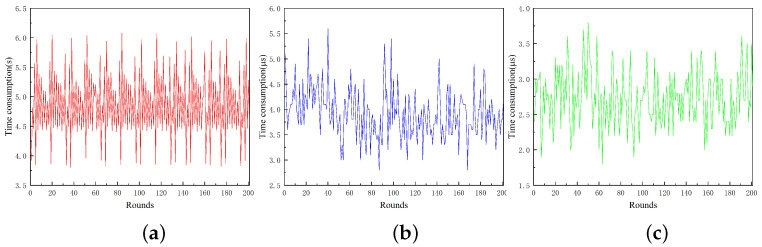
The Effect of location order on the performance of GL-Tree in location dataset. (**a**) Creation GL-Tree. (**b**) Location insertion. (**c**) Location retrieval.

**Table 1 sensors-23-02245-t001:** Explanation of the process of coding the latitude.

Latitude	Rounds	Interval	Interval (Code 0)	Interval (Code 1)	Encode
	1	(−90, 90)	(−90, 0)	(0, 90)	1
	2	(0, 90)	(0, 45)	(45, 90)	0
	3	(0, 45)	(0, 22.5)	(22.5, 45)	1
	4	(22.5, 45)	(22.5, 33.75)	(33.75, 45)	1
39.965391	5	(33.75, 45)	(33.75, 39.375)	(39.375, 45)	1
	6	(39.375, 45)	(39.375, 42.1875)	(42.1875, 45)	0
	7	(39.375, 42.1875)	(39.375, 40.78125)	(40.78125, 42.1875)	0
	...	...	...	...	...
	20	(39.96517, 39.96552)	(39.96517, 39.96534)	(39.96534, 39.96552)	1

**Table 2 sensors-23-02245-t002:** Geohash code precision variation.

Length of Geohash Code	Latitude Bit	Longitude Bit	Latitude Error	Longitude Error	Height	Width
1	2	3	±23	±23	4992.6 km	5009.4 km
2	5	5	±2.8	±5.6	624.1 km	1252.3 km
3	7	8	±0.70	±0.70	156 km	156.5 km
4	10	10	±0.087	±0.18	19.5 km	39.1 km
5	12	13	±0.022	±0.022	4.9 km	4.9 km
6	15	15	±0.0027	±0.0055	609.4 m	1.2 km
7	17	18	±0.00068	±0.00068	152.5 m	152.9 m
8	20	20	±0.00086	±0.000172	19 m	38.2 m
9	22	23	±0.000021	±0.000021	4.8 m	4.8 m

## Data Availability

Not applicable.

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
