# Peer review of "GL-Tree: A Hierarchical Tree Structure for Efficient Retrieval of Massive Geographic Locations"

_sensors, 2023, doi:10.3390/s23042245_

Round 1

Reviewer 1 Report

This is a very timely paper.

Author Response

Dear Reviewers.

  Thank you very much for your review of this article with the other three reviewers. It has been revised based on all the review comments. The specific revisions are described as follows.

  1. A description of the experimental results has been added to the end of the abstract.
  2. In Section 1.INTRODUCTION. we has been revised to clarify the motivation of this paper based on the description and problem analysis of the centralized location privacy protection structure and to adjust this section into four subsections.

1.1.overview

1.2. research objectives and motivation

1.3. proposal overview

1.4. paper outline

    3.In section 2.related work. we has been modified to add the study of work related to location privacy protection, and this section has been adjusted into 4 subsections based on the relationship between the contents。

2.1. Research related to location privacy protection

2.2. Spatial indexing techniques in location-based applications

2.3. Application of Spatial Indexing Techniques in Location Privacy Protection

2.4. Comparative Analysis

And in section 2.4, the problem of different tree index structures has been added to analyze the reasons for using GL-Tree and Geohash code in this study

     4.In 6. Experiments and Results Analysis . The experimental platform and dataset size were adjusted according to the review experts' opinions. The experimental platform was adjusted from the original DELL laptop, WINDOWS 10 OS, to Aliyun elastic computing server, windows server2019 datacenter OS. the dataset size was adjusted from 500000 to 1250000, and the experiments and experimental data analysis were conducted again.

    5.In 7. Conclusions. For future work ideas, were adjusted to add the research idea of combining blockchain technology, differential privacy technology, and GL-Tree.

  1. Adjusted some long sentences and grammar in the article.

The above is a description of the changes made based on all the reviewers' comments.

Thank you again for all your hard work. I wish you a happy New Year!

                                                                     Kind regards

                                                                      The author

Reviewer 2 Report

The paper is well structured. It is little difficult to interpret because of the highly-technical terms but the manuscript can be followed by the reader. Please, this reviewer suggests thoroughly checking the grammar and spelling. As an example, in line rows 496, too much long sentences are used where the meaning is lost, "In location privacy preserving".   

In abstract, how the results relate with the experiments done is missing? A line can be added.  

The author can mention the objectives precisely by adding a separate subsection on motivation and contributions under Introduction section. The references are adequate and reflect the state of the art. The results are well described and the conclusions are in accordance with what has been done. The authors can also mention explicitly the  limitations of their work  

There is small doubt, "Mentioned in row 565, the tree stores all locations within this small interval, how the data is uniformly stored. Please explain ? What is the expected load in this methodology with respect to storage?

Author Response

Dear Reviewer.

  Thank you very much for your review of this article with the other three reviewers. It has been revised based on all the review comments. The specific revisions are described as follows.

Response to your valuable suggestions:

Suggestion 1: In abstract, how the results relate with the experiments done is missing? A line can be added.

Response :A description of the experimental results has been added to the end of the abstract.

Suggestion 2:The author can mention the objectives precisely by adding a separate subsection on motivation and contributions under Introduction section. The references are adequate and reflect the state of the art. The results are well described and the conclusions are in accordance with what has been done. The authors can also mention explicitly the  limitations of their work  

Response In Section 1.INTRODUCTION . we has been revised to clarify the motivation of this paper based on the description and problem analysis of the centralized location privacy protection structure and to adjust this section into four subsections.

1.1.overview

1.2. research objectives and motivation

1.3. proposal overview

1.4. paper outline

In section 2.related work. we has been modified to add the study of work related to location privacy protection, and this section has been adjusted into 4 subsections based on the relationship between the contents。

2.1. Research related to location privacy protection

2.2. Spatial indexing techniques in location-based applications

2.3. Application of Spatial Indexing Techniques in Location Privacy Protection

2.4. Comparative Analysis

And in section 2.4, the problem of different tree index structures has been added to analyze the reasons for using GL-Tree and Geohash code in this study

Suggestion 3:There is small doubt, "Mentioned in row 565, the tree stores all locations within this small interval, how the data is uniformly stored. Please explain ? What is the expected load in this methodology with respect to storage?

Response: Additional explanation has been presented in the paper, and the adjustment is mentioned in line 725.

Suggestion 4:this reviewer suggests thoroughly checking the grammar and spelling.。

Response:Adjusted some long sentences and grammar in the article.

The above is a description of the changes made based on your comments. Thank you again for all your hard work. I wish you a happy New Year!

                                                                                          Kind regards

                                                                                            The author

Reviewer 3 Report

The paper designed a hierarchical tree-like organization structure, GL-Tree, which enables the storage, management, and retrieval of massive location data and satisfies the user’s location-hiding requirements by using Geohash encoding to convert the two-dimensional geospatial coordinates of locations into one-dimensional strings and construct the GL-Tree based on the Geohash encoding principle.

The paper is interesting, but still need more enhancements by performing the following comments:

First, the authors must include the main results in the abstract and compared it with other previous works in briefly.

Second, the authors must clarify the main objectives and motivation of the study in the Introduction section. 

Third, In related works, the authors did not mention any other types of common algorithms that are used for location-based application services and location privacy protection. Some examples include:

1- k-anonymity: This algorithm is used to protect the privacy of individuals by obscuring their location data in a group of at least k other individuals.

2- Differential privacy: This algorithm is used to protect the privacy of individuals by adding noise to location data before it is shared with a third party.

3- Spatial cloaking: This algorithm is used to protect the privacy of individuals by obscuring their location data within a certain radius.

4- Location-based encryption: This algorithm is used to protect the privacy of location data by encrypting it before it is shared with a third party.

5-Geo-indistinguishability: This algorithm is used to protect the privacy of individuals by obscuring their location data within a certain area.

6-Spatial hashing: This algorithm is used to protect the privacy of location data by representing it as a hash value before it is shared with a third party.

So I request from authors to compare between them and why use GL-Tree and Geohash code. 

Fourth, cite the following studies related to the common security issues in different domains:

- Classification of cyber security threats on mobile devices and applications.

- Improved security particle swarm optimization (PSO) algorithm to detect radio jamming attacks in mobile networks.

- A new hybrid text encryption approach over mobile ad hoc network

Fifth, I suggest for authors to add future work to deploy blockchain technology  for location-based application services and location privacy protection. You can cite the following study:

- A new scheme for detecting malicious attacks in wireless sensor networks based on blockchain technology

My decision is Major revision. The paper can be accepted after doing the above comments.

Author Response

Dear Reviewer.

  Thank you very much for your review of this article with the other three reviewers. It has been revised based on all the review comments. The specific revisions are described as follows.

Response to your valuable suggestions:

Suggestion 1: First, the authors must include the main results in the abstract and compared it with other previous works in briefly.

Response :A description of the experimental results has been added to the end of the abstract.

Suggestion 2:Second, the authors must clarify the main objectives and motivation of the study in the Introduction section.

Response :In Section 1.INTRODUCTION . we has been revised to clarify the motivation of this paper based on the description and problem analysis of the centralized location privacy protection structure and to adjust this section into four subsections.

1.1.overview

1.2. research objectives and motivation

1.3. proposal overview

1.4. paper outline

Suggestion3: Third, In related works, the authors did not mention any other types of common algorithms that are used for location-based application services and location privacy protection. So I request from authors to compare between them and why use GL-Tree and Geohash code. 

Response :In section 2.related work. we has been modified to add the study of work related to location privacy protection, and this section has been adjusted into 4 subsections based on the relationship between the contents。

2.1. Research related to location privacy protection

2.2. Spatial indexing techniques in location-based applications

2.3. Application of Spatial Indexing Techniques in Location Privacy Protection

2.4. Comparative Analysis

And in section 2.4, the problem of different tree index structures has been added to analyze the reasons for using GL-Tree and Geohash code in this study

Suggestion4: Fourth, cite the following studies related to the common security issues in different domains:

Response :First of all, thank you very much for this valuable comment. The focus of this paper is to discuss the problem of storing and retrieving GL-Tree in massive location sets and discuss its application in location privacy protection. You suggested supplementing the research content of different fields. We have read the related papers carefully, but we feel that they are related to this paper but not very relevant. Adding them to related research would have an impact on the logic of the paper. At the same time, many related studies have been described and the number of references has reached 40. Therefore, we would like to discuss with you further whether it is necessary to add these contents.

Suggestion5:Fifth, I suggest for authors to add future work to deploy blockchain technology  for location-based application services and location privacy protection.  

Response :In 7. Conclusions. For future work ideas, were adjusted to add the research idea of combining blockchain technology, differential privacy technology, and GL-Tree.

Other Modification

  1. In 6. Experiments and Results Analysis . The experimental platform and dataset size were adjusted according to the review experts' opinions. The experimental platform was adjusted from the original DELL laptop, WINDOWS 10 OS, to Aliyun elastic computing server, windows server2019 datacenter OS. the dataset size was adjusted from 500000 to 1250000, and the experiments and experimental data analysis were conducted again.
  2. Adjusted some long sentences and grammar in the article.

The above is a description of the changes made based on all the reviewers' comments. Thank you again for all your hard work. I wish you a happy New Year!

                                                                            Kind regards

                                                                              The author

Reviewer 4 Report

In this paper, a tree structure is proposed for geolocation retrieval. The proposed method combines the geocode with L-tree, to create a framework to manage the geolocation data. This method is evaluated with the dataset and compared with the classical B+ tree and R tree. The results indicate that the proposed method is over-performance than the classical ones.
However, the method is not compared with the start of art methods which makes it difficult to evaluate its performance. Furthermore, the size of test dataset is too small and the tasks (create, insert, retievel) are basic operations which may different from some real applications. Therefore, larger test dataset and different application results should be proved. Another consideration is about the computation platform. The test is running in a PC which is currently replace by cloud platform to geolocation process. How about the application of the proposed method in cloud platform should be discussed with experiment results.

Overall, this paper is interesting for the readers but more experimental results should be provided before it can be further considered for the publication.

Author Response

Dear Reviewer.

  Thank you very much for your review of this article with the other three reviewers. It has been revised based on all the review comments. The specific revisions are described as follows.

Response to your valuable suggestions:

First of all, thank you very much for your valuable suggestions. Regarding the experimental part, we have followed your suggestion to migrate the experiment from laptop to Ali cloud server and increase the dataset of the experiment. The details are described below.

   In 6. Experiments and Results Analysis . The experimental platform and dataset size were adjusted according to the review experts' opinions. The experimental platform was adjusted from the original DELL laptop, WINDOWS 10 OS, to Aliyun elastic computing server, windows server2019 datacenter OS. the dataset size was adjusted from 500000 to 1250000, and the experiments and experimental data analysis were conducted again.

The focus of this paper is to discuss the problem of storing and retrieving GL-Tree in massive location sets and discuss its application in location privacy protection. The additional experiments you propose are very reasonable. This is what we are currently working on. The specific work is to design a GL-Tree location privacy protection scheme based on the nearest neighbor query and spatial range query as application scenarios and perform performance tests. Related research is in progress and we intend to discuss it in the next paper. We hope to receive your approval. Thank you very much.

Other Modification:

  1. A description of the experimental results has been added to the end of the abstract.
  2. In Section 1.INTRODUCTION. we has been revised to clarify the motivation of this paper based on the description and problem analysis of the centralized location privacy protection structure and to adjust this section into four subsections.

1.1.overview

1.2. research objectives and motivation

1.3. proposal overview

1.4. paper outline

3.In section 2.related work. we has been modified to add the study of work related to location privacy protection, and this section has been adjusted into 4 subsections based on the relationship between the contents。

2.1. Research related to location privacy protection

2.2. Spatial indexing techniques in location-based applications

2.3. Application of Spatial Indexing Techniques in Location Privacy Protection

2.4. Comparative Analysis

And in section 2.4, the problem of different tree index structures has been added to analyze the reasons for using GL-Tree and Geohash code in this study

4.In 7. Conclusions. For future work ideas, were adjusted to add the research idea of combining blockchain technology, differential privacy technology, and GL-Tree.

5.Adjusted some long sentences and grammar in the article.

The above is a description of the changes made based on all the reviewers' comments. Thank you again for all your hard work. I wish you a happy New Year!

                                                                                    Kind regards

                                                                                      The author

Round 2

Reviewer 3 Report

Some of comments did not achieve 

Author Response

Dear Reviewer.

Thank you very much for your review of this article. 

Based on the first round of revisions, in response to your suggestion to supplement the research on security issues in different fields, we have added and improved in the first section INTRODUCTION, citing the four papers you suggested (reference ID: 2,4,7,8) and other literature. We have also adjusted the content of the INTRODUCTION section and its organizational order to make it more logically related.

In addition, we revised the English language expression, grammar, and format of the whole text, paragraph by paragraph.

Now, we have responded to all of your suggestions. Please review.

Thank you again for all your hard work. 

                                                                            Kind regards

                                                                              The author

Reviewer 4 Report

The paper has been revised according to the comments from the reviewer. It is suggested to accept the paper for publication after some minor checks on languages and format as required by the editors.

Author Response

Dear Reviewer.

Thank you very much for your review of this article. 

According to your suggestion, we revised the English language expression, grammar, and format of the whole text, paragraph by paragraph.

Thank you again for all your hard work. 

                                                                            Kind regards

                                                                              The author